# Differential Diagnosis of OKC and SBC on Panoramic Radiographs: Leveraging Deep Learning Algorithms

**DOI:** 10.3390/diagnostics14111144

**Published:** 2024-05-30

**Authors:** Su-Yi Sim, JaeJoon Hwang, Jihye Ryu, Hyeonjin Kim, Eun-Jung Kim, Jae-Yeol Lee

**Affiliations:** 1Department of Oral and Maxillofacial Surgery, Dental and Life Science Institute & Dental Research Institute, School of Dentistry, Pusan National University, Yangsan 50612, Republic of Korea; la_vender@pusan.ac.kr (S.-Y.S.); ryujh@umich.edu (J.R.); twdt92@gmail.com (H.K.); 2Department of Oral and Maxillofacial Radiology, Dental and Life Science Institute & Dental Research Institute, School of Dentistry, Pusan National University, Yangsan 50612, Republic of Korea; softdent@pusan.ac.kr; 3Department of Dental Anesthesia and Pain Medicine, School of Dentistry, Pusan National University, Yangsan 50612, Republic of Korea; anekej@pusan.ac.kr

**Keywords:** deep learning, differentiation, diagnosis, odontogenic keratocyst (OKC), simple bone cyst (SBC)

## Abstract

This study aims to determine whether it can distinguish odontogenic keratocyst (OKC) and simple bone cyst (SBC) based solely on preoperative panoramic radiographs through a deep learning algorithm. (1) Methods: We conducted a retrospective analysis of patient data from January 2018 to December 2022 at Pusan National University Dental Hospital. This study included 63 cases of OKC confirmed by histological examination after surgical excision and 125 cases of SBC that underwent surgical curettage. All panoramic radiographs were obtained utilizing the Proline XC system (Planmeca Co., Helsinki, Finland), which already had diagnostic data on them. The panoramic images were cut into 299 × 299 cropped sizes and divided into 80% training and 20% validation data sets for 5-fold cross-validation. Inception-ResNet-V2 system was adopted to train for OKC and SBC discrimination. (2) Results: The classification network for diagnostic performance evaluation achieved 0.829 accuracy, 0.800 precision, 0.615 recall, and a 0.695 F1 score. (4) Conclusions: The deep learning algorithm demonstrated notable accuracy in distinguishing OKC from SBC, facilitated by CAM visualization. This progress is expected to become an essential resource for clinicians, improving diagnostic and treatment outcomes.

## 1. Introduction

Machine learning is the most significant advancement in artificial intelligence (AI). AI learns from algorithms, requiring data to learn patterns to make decisions or predictions based on the interpretation of data without specific programming [1]. In machine learning, building a training dataset with sufficient information and accurately categorizing the data are the most important first steps. The tagged data is pre-processed to guarantee appropriate training steps and exported in the right format. On the contrary, deep learning develops predictions through autonomically perceived data rather than handcrafted data. A convolutional neural network (CNN) architecture in two and three dimensions is used for deep learning, with different training steps implemented. Multi-layered artificial neural networks ensure that the system automatically accumulates and turns this data into output by easily detecting the contours of the structure to be presented to the system [2].

Deep learning has been developed rapidly in medical radiographs [3], including the detection of abnormalities, classification, separation of lesions or organs, measurement of volume or length, and image transformation or reconstruction. Research has been conducted on the use of deep learning methods in various fields of the dental profession [4]. Many of them were related to dental morphology, other subjects such as gingival recession, osteoporosis, and anatomical structures like inferior alveolar canals were also studied through deep learning [5]. An especially deep learning algorithm has been implemented for the detection of landmarks in cephalometric radiographs [6] and the diagnosis of diseases like caries detection [7], periodontal bone loss [8], and periapical lesions [9].

Panoramic radiographs are a standard imaging modality and a typical diagnostic tool that dentists often use in their routine clinical practice when patients first visit a dental clinic, as well as for follow-up [10]. Panoramic radiographs are preferred because they allow the complete dentition and surrounding anatomical landmarks to be assessed with a single radiation exposure [11].

When a patient first visits a dental clinic, regardless of the patient’s chief complaint, clinicians often accidentally discover cystic lesions on panoramic radiographs that do not cause any symptoms [12]. Benign cysts and tumors in maxillofacial bones are usually painless and asymptomatic unless they have expanded to the point where they affect significant parts of the bone, for example, a pathological fracture caused by expanded or thinned cortical bone [13]. Some intrabony lesions are asymptomatic in the early stages, a delayed diagnosis and a poor prognosis are likely to follow when they are misdiagnosed [14,15]. It is imperative to conduct a comprehensive analysis and diagnosis of the lesion to determine the most suitable treatment.

Given that most cysts and tumors typically manifest on radiographs as radiolucent lesions with corticated margins, establishing a precise differential diagnosis can pose challenges. [16]. Odontogenic keratocyst (OKC) and simple bone cyst (SBC) exhibit similar characteristics on panoramic radiographs yet necessitate different treatments and result in distinct prognoses. SBC mostly affects the mandible and is typically asymptomatic. It is more prevalent during the second decade of life [17]. Conversely, OKC predominantly affects the mandible in a 2:1 ratio to the maxilla and occurs during the second and third decades of life [18,19]. OKC demonstrates a biological pattern like that of a benign tumor, necessitating surgical excision for treatment, and is associated with a high recurrence rate [20]. In contrast, SBC tends to be self-limiting, with spontaneous resolution occurring frequently, sometimes eliminating the need for surgical intervention [17,21]. Due to the similarities of SBC to OKC on panoramic radiographs, its differential diagnosis of SBC requires additional cone beam computed tomography (CBCT) imaging and sometimes biopsy. This not only consumes more time in the examination process but also poses potential risks to patients, including unnecessary radiation exposure and surgical procedures, which are typically not warranted [22].

OKC shows a biological pattern like that of a benign tumor, requires surgical excision as a treatment method, and has a high recurrence rate [23], while SBC is self-limiting [24]. 

Considering these facts, the competence of clinicians is crucial in determining both the patient’s prognosis and the selection of appropriate treatment. Particularly, in the field of dentistry, individual knowledge, expertise, and biases significantly influence the agreement rate (a proxy for diagnostic performance) between dental practitioners’ interpretations of radiographs [25]. 

Therefore, this study aims to determine whether a deep learning algorithm can effectively differentiate between OKC and SBC, two lesions with similar radiographic characteristics, solely based on preoperative panoramic radiographs.

## 2. Materials and Methods

The ethics committee of Pusan National University Dental Hospital approved this study (No. 2024-03-010-004), which followed the principles outlined in the Declaration of Helsinki. Since the study was retrospective, informed consent was not required.

Patients with SBC and OKC who visited Pusan National University Dental Hospital between January 2018 and December 2022 had their panoramic radiographs taken. All panoramic radiographs were obtained utilizing the Proline XC system (Planmeca Co., Helsinki, Finland), which already had diagnostic data on them. CBCT was used to support the preliminary diagnosis of SBC and OKC. This study included 63 cases of OKC confirmed by histological examination after surgical excision and 125 cases of SBC that underwent surgical curettage (Figure 1). All 63 OKC cases were confirmed by histopathological examination after surgical excision. The 125 SBC cases underwent curettage because the cavity was empty and often contained connective tissue, so a biopsy was unnecessary. Periodic follow-ups were performed for both OKC and SBC patients before they were discharged. All cases were in the mandible, not associated with impacted teeth, and panoramic radiographs were acquired using a single imaging system. Only images with clear depictions of well-defined circumscribed lesions were included. Images were preprocessed by cropping the region of interest (ROI) to improve the prediction accuracy of the deep learning system. The processed panoramic images were analyzed using the Inception-ResNet-V2 [26] network framework to learn the lesion’s morphology and obtain an algorithm. The accuracy and specificity of lesion identification were then analyzed. 

### 2.1. Data Preprocessing

We converted 2943 × 1435 panoramic images into learnable data [27]. Panoramic images with histopathological diagnoses of OKC and SBC were converted to ground truth images and labeled using MATLAB 2023b (MathWorks, Inc., Natick, MA, USA). The ROI was retrieved from panorama images using element-wise augmentation and cropped into a square form scaled to 299 × 299 to retain the horizontal and vertical ratios (Figure 2).

### 2.2. Image Augmentation

To prevent overfitting due to the insufficient amount of data, we increased the number of training images by rotating them from −10 to 10 degrees and translating them from −5 to 5 pixels horizontally and vertically [28]. This work was conducted using a workstation (i7-8700K CPU, 32GB RAM) with an NVIDIA Titan RTX GPU(NVIDA, CA, USA).

### 2.3. Data Preperation Using 5-Fold Cross-Validtion

After preprocessing, we created five sets for training the AI deep learning algorithm, with each set containing 204 images categorized into two groups for 5-fold cross-validation. The dataset was divided into training and validation sets, yielding 163 images for the training group and 41 images for the validation group.

We used 5-fold cross-validation to prevent bias due to the limited quantity of data used to train the deep learning models for image classification. The validation set was reserved for the final evaluation of the model’s performance and used only once. The training set was then split into five equal parts. During the cross-validation process, each part was used as a validation set once, with the remaining four parts forming the new training set. This ensured that every data point was used for both training and validation exactly once. The network was trained with these images, and its performance was validated.

### 2.4. Network Architecture

We used Inception-ResNet-V2 [26] network(see Table 1), pretrained on over a million images from the ImageNet archive, allowing the deep learning algorithms to learn a wide range of images. To reduce diagnostic mistakes resulting from class imbalance, inverse-frequency class weights were implemented since the data ratio was OKC:SBC = 1:2.

### 2.5. Training Options

We used a workstation (i7-8700K CPU, 32GB RAM) with an NVIDIA Titan RTX GPU for training the networks using MATLAB 2023b (MathWorks, Natick, MA, USA). 

Models were trained using the Adam optimizer for a maximum of 500 epochs [29]. The initial learning rate was 10^−4^, and the mini-batch size was 16. Early stopping of the training procedure occurred if validation accuracy did not increase for more than 30 epochs to prevent overfitting.

### 2.6. Accuracy and Precision Assessment

The accuracy and precision assessment and time analysis were performed in two groups: the model and human clinicians. The human clinicians consisted of an OMR specialist and three general practitioners (GPs). The Oral and Maxillofacial Radiology Specialists (OMR) have over 10 years of experience in diagnostic interpretation. The general practitioners (GPs) are interns at Pusan National University Dental Hospital who have passed the dental board test but have not yet started specialized training. Accuracy, precision, recall, and F1 score were defined from the confusion matrix. True positive (TP), false negative (FN), true negative (TN), and false positive (FP) metrics were used to calculate accuracy and F1 scores.
$$Accuracy=\frac{TP+TN}{TP+FP+FN+TN}$$
$$Precision=\frac{TP}{TP+FP}$$
$$Recall=\frac{TP}{TP+FN}$$
$$F1\;Score=\frac{2\times (Recall\times Precision)}{Recall+Precision}$$

TP: true positive, FP: false positive, FN: false negative, TN: true negative

### 2.7. Visualizing Model Decision

Class Activation Mapping (CAM) is a method that removes the fully connected layer from CNN while preserving location information, allowing visualization of which part of the image AI is classifying [30]. A gradient-based heat map was created as a guide to highlight the important regions of the class label on the image in a reddish color [31]. 

## 3. Results 

The trained model showed 0.829 accuracy, 0.800 precision, 0.615 recall, and 0.695 F1 score. Human performance is estimated in Table 2. The diagnostic accuracy of the model resulted in 0.805, and OMR specialists showed similar accuracy (0.805), while GP 1, 2, and 3 showed lower results (0.439, 0.707, and 0.463). The model demonstrated immediate detection and classification competence, while human physicians (OMR specialists and GPs) took an average of 415 s to analyze all 41 validation set images.

### 3.1. A Confusion Chart

We calculated the accuracy using elements of the confusion matrix (Figure 3). For OKC (Group 1), the classifier correctly identified 26 cases but misclassified 2 cases as SBC. For SBC (Group 2), the classifier correctly identified eight instances but misclassified five instances as OKC.

### 3.2. Visualization of Model Classification

In Figure 4a, AI was classified as SBC with a probability of 99.46%, whereas in Figure 4b, OKC was classified as OKC with a probability of 69.28%. Upon comparison of the CAM images of SBC and OKC, it was evident that the interior of the OKC lesions exhibited a notably redder appearance. Additionally, our analysis identified that the CAM hot spots were situated at or near the lesion’s periphery. In Figure 4c, OKC was misclassified as SBC with a probability of 87.14%, and in Figure 4d, SBC was misclassified as OKC with a probability of 75.84%. In both of these images, the CAM hot spots are looking at unrelated areas, not at the lesions.

### 3.3. Cross-Tabulation of OMR Specialists and AI Results

To further evaluate whether the AI mimics the human decision process or employs a different strategy, we conducted a cross-tabulation of the diagnostic results of the AI model against those of the OMR specialist. The comparison is summarized in Figure 5. The AI model and the OMR specialist agreed on 26 cases of OKC and 8 cases of SBC. However, the AI model identified five cases as OKC that the OMR specialist identified as SBC, and two cases as SBC that the OMR specialist identified as OKC.

### 3.4. Confidence Levels of AI Model Predictions 

The confidence levels provided by the AI model are crucial for assessing the reliability of its diagnostic predictions. The distribution of confidence levels for correct and incorrect classifications is summarized in Figure 6. Most predictions are highly confident, with 32 predictions above 0.95 and only a few predictions below 0.8.

## 4. Discussion

Clinicians identify diverse lesions, from salivary stones and cysts to benign tumors and malignancies, through panoramic radiographs, minimizing patient radiation exposure [32,33]. However, distinguishing between OKC and SBC is challenging for clinicians due to their similar appearance on panoramic radiographs. Both may appear as well-defined radiolucent lesions in the posterior mandible with scalloped margins [16]. Misdiagnosing SBC as OKC may lead to unnecessary surgical interventions for patients, especially when SBC typically necessitates conservative follow-up care. Conversely, misidentifying OKC as SBC may result in missing the optimal time for surgical intervention or delayed detection of recurrence. Hence, differential diagnosis holds significance as the treatment modalities and prognostic outcomes differ between the two lesions.

Deep learning algorithms can be beneficial for clinicians to make more precise diagnostic impressions. Birdal et al. reported the efficient use of various automatic diagnostic methods for panoramic image interpretation as an assistant for examination [34]. Compared to manual segmentation, a deep learning-based automatic method showed faster and more precise performance in detecting and segmenting subjects on panoramic radiographs [35]. Ari et al. reported that the present algorithm is well-trained to reliably distinguish SBC with excellent accuracy in panoramic images from odontogenic cysts or tumors [34]. Several studies have demonstrated that deep learning algorithms have proven to be able to differentiate between OKC and ameloblastoma with sufficient accuracy. This can assist surgeons in determining appropriate treatment strategies and intervening at the appropriate time [36,37]. 

The analysis of preoperative panoramic radiographs using a deep learning algorithm showed reliable accuracy in distinguishing between OKC and SBC. The ROI classification was established, incorporating augmentation through random rotation between 0 and 10 degrees. This implies that the inclusion of a rotational parameter likely contributed to the development of a more efficient deep learning model by enhancing data representation across diverse cases [38]. Our data showed reliable classification performance, with an accuracy of 82.9%. Due to the prevalence of SBC cases, we assigned additional class weight to OKC to counterbalance the random selection bias towards SBC. The confusion matrix shows that the deep learning algorithm attained a higher accuracy rate in predicting OKC cases. 

The cross-tabulation (Figure 5) of the AI model’s results with those of the OMR specialist shows a high level of agreement. To quantify the level of agreement between the AI model and the OMR specialist, we calculated Cohen’s Kappa coefficient, which measures inter-rater agreement for categorical items. The calculated Kappa value (*κ*κ) was approximately 0.58, indicating a moderate to substantial agreement between the AI model and the OMR specialist [39]. This high level of agreement suggests that the AI model has learned to identify similar patterns to those recognized by experienced human diagnosticians, indicating that it mimics the human decision-making process to a considerable extent.

Figure 3 shows that CAM highlights the border region between the lesions and teeth, emphasizing crucial structures like the tooth root and parts of the lesion’s margin. Typically, a periodontal ligament (PDL) space is located at the lesion’s border. OKC is sometimes associated with nonvital teeth [40,41], while SBC usually corresponds with vital teeth [42]. In our study, most SBC cases exhibited normal PDL space where teeth were adjacent to the lesion’s margin.

The density and depth of cysts serve as additional features for lesion characterization, providing valuable information for differential diagnosis. O. Ferreira Ju’nior et al. examined the pixel gray levels within the cysts of OKC and SBC. The varying gray levels observed in OKC indicate denser content within the cyst compared to the empty or partially empty cavity of SBC. Pixel values inside OKC were found to be higher than those in SBC images, with these values closely associated with the density of the lesion contents. [43]. Similarly, when compared to the CAM of SBC, the CAM for OKC demonstrates the lesion interior being significantly redder. The information regarding the contours of OKCs and SBCs, along with the pixel gray level representing the density of their fluid content, proved to be valuable in distinguishing between these lesions [44]. In our results, we also observed that hot spots of the CAM were partially located around the margins of the lesion. This knowledge provides the deep learning system with an indicator as to how to differentiate between OKC and SBC. With the collection of more data in the future, the regions highlighted by CAM will become more discernible for clinicians, thereby facilitating easier differentiation between them. Utilizing CAM, the provided images of highlighted lesions may expedite the detection of characteristic features of these lesions on panoramic radiographs, potentially obviating the need for additional CBCT scans.

This study highlights the potential of the deep learning model to reduce the reliance on CBCT scans. Deep learning algorithms can be beneficial for clinicians, especially general practitioners. The OMR acquired as much high accuracy and precision as the deep learning algorithm, but the GP showed comparably low accuracy in making differential diagnoses, possibly leading to misdiagnosis results. However, given the current predictive values, the approach may not completely eliminate the need for CBCT in all cases. Instead, it could serve as a robust preliminary diagnostic tool to identify cases where further imaging is most necessary, thus optimizing the use of CBCT and reducing unnecessary exposure for patients. 

The confidence levels provided by the AI model are crucial in assessing the reliability of its diagnostic predictions (Figure 6). The model outputs a probability score for each classification, reflecting its confidence level. In our study, the AI model achieved a substantial confidence level in its predictions, with a mean probability score of 0.985 for correct classifications. Notably, 87% of the predictions had confidence levels exceeding 95%, supporting the reliability of the AI model for initial diagnostic purposes.

To integrate this approach more concretely into the diagnostic pipeline, a tiered diagnostic strategy is proposed. Initially, panoramic radiographs can be analyzed using the deep learning model to identify potential OKC or SBC cases. Cases with high confidence levels in the AI predictions may proceed with treatment based on these findings, while cases with lower confidence or ambiguous results can be referred for CBCT imaging for further evaluation. This approach leverages the strengths of both modalities and ensures a balance between diagnostic accuracy and patient safety. The high proportion of predictions with confidence levels above 95%, as shown in Figure 6, validates the proposed tiered diagnostic strategy and supports its implementation.

This study was limited by the inability to use a large amount of clinical data due to constraints, potential biases introduced by the retrospective design, and the limitations of a single-center study environment. To mitigate these biases and limitations, future studies should consider prospective designs and multicenter collaborations. Multicenter trials can be more explainable and supportive of the study results, providing larger sample sizes that could be more broadly applied and generalizable. Collecting more data in collaboration with other hospitals will yield more accurate results.

Although panoramic radiographs alone are important for clinicians to make a provisional diagnosis, image distortion may occur in panoramic radiographs when X-rays penetrate the cyst wall at an oblique angle, causing the focal layer to be narrower in the anterior region. The mandibular thickness, the patient’s position, and the ghost image of the spine may also be attributed to this distortion [16].

As a complement, CBCT serves as an excellent imaging modality for the maxillofacial region, offering clear, highly contrasted images of anatomical structures and a valuable tool for bone evaluation [44,45]. Deep learning combining CBCT images also enhanced the quality of radiograph interpretation. For example, a study on airway volume measurement on CBCT radiographs shows that automatic segmentation trained by a deep learning model is proven to be reliable [46]. Several studies have investigated the boundaries of OKC and SBC on CBCT images. Specifically, in OKC, the sclerotic border of the lesion was found to be more prevalent, especially in the posterior region. A scalloping appearance can also be seen on the superior border of the SBC. Jiang et al. emphasized that SBC showed a scalloped margin on the panoramic radiograph, with the upper boundary of the cyst extending to the tooth root. Additionally, they noted that, on CBCT imaging, the cyst appears fan-shaped with distinct hard bone lines, resembling the shape of a shell [47]. They also observed that the rate of cortical bone perforation in OKC was 36.8%, with bone thinning present in 89.5% of cases. In SBC cases, only cortical bone thinning was noted, with no instances of perforation [47]. CBCT also offers clear imaging advantages [48], supports superior dental diagnoses with deep learning [9,49,50] and automates lesion detection to reduce radiologists’ workloads. Therefore, for improved accuracy and precision, we intend to pursue a future study to incorporating the use of CBCT. This can be achieved by integrating a deep learning algorithm with additional three-dimensional information provided by CBCT scans. 

## 5. Conclusions

This study highlights the utility of panoramic radiographs in the initial diagnosis of oral lesions, emphasizing the minimal radiation exposure benefit. Distinguishing between OKC and SBC presents a challenge due to their similar appearances on radiographs, where accurate differentiation is crucial for appropriate management. Deep learning algorithms have proven effective in enhancing diagnostic precision, and leveraging lesion characteristics to improve differentiation. While our findings indicate the potential of deep learning to streamline the diagnostic process on radiographs and reduce the reliance on additional CBCT scans, limitations in clinical data underscore the need for further research. Incorporating CBCT with deep learning for future studies appears promising for advancing diagnostic accuracy and patient care in oral lesion management.

## Figures and Tables

**Figure 1 diagnostics-14-01144-f001:**
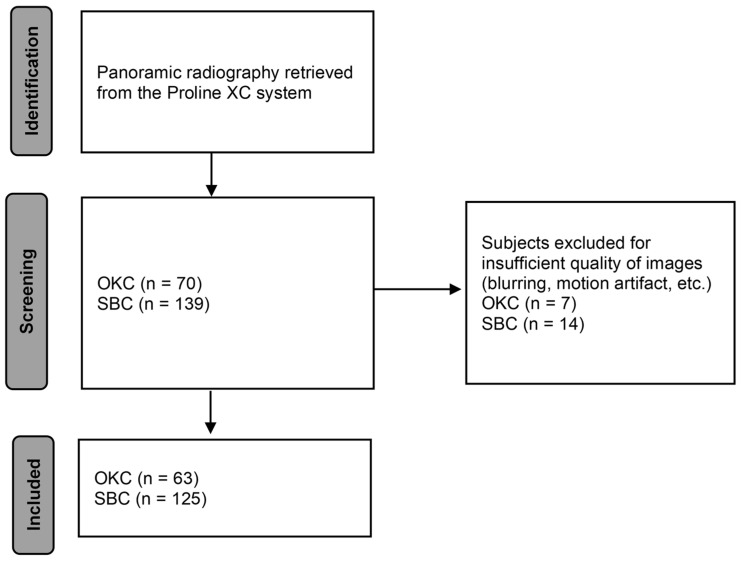
Flow diagram for identification of subjects and inclusions.

**Figure 2 diagnostics-14-01144-f002:**
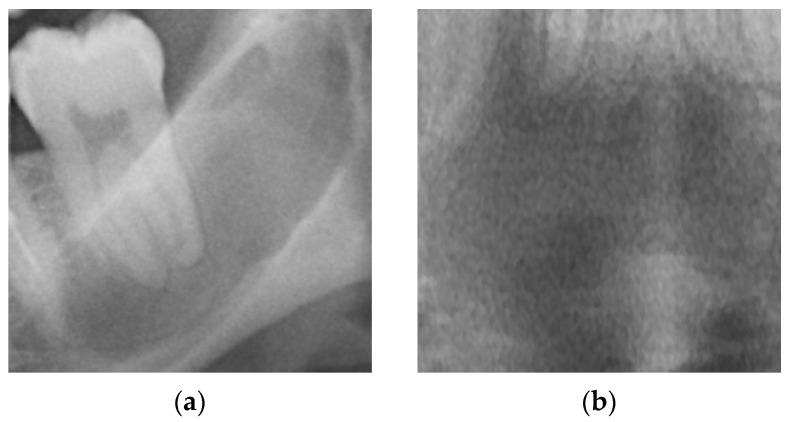
Two examples of manually cropped images of panoramic radiography: (**a**) OKC, (**b**) SBC.

**Figure 3 diagnostics-14-01144-f003:**
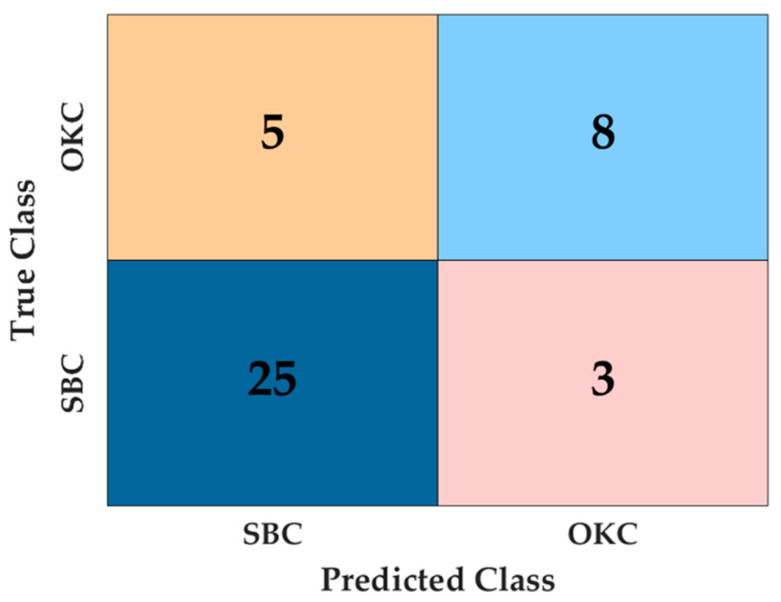
A confusion matrix of OKC classification from SBC based on test data. Groups 1 and 2 were set to OKC and SBC, respectively.

**Figure 4 diagnostics-14-01144-f004:**
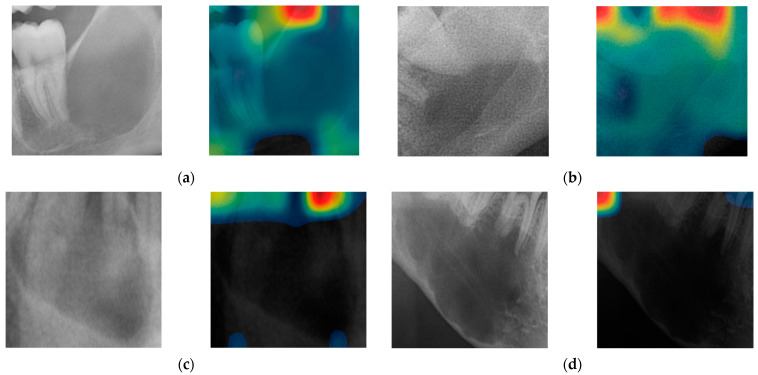
Four examples of cropped images alongside their corresponding class activation maps (CAM): correctly classified cases: (**a**) SBC and (**b**) OKC; incorrectly classified cases: (**c**) OKC misclassified as SBC, and (**d**) SBC misclassified as OKC.

**Figure 5 diagnostics-14-01144-f005:**
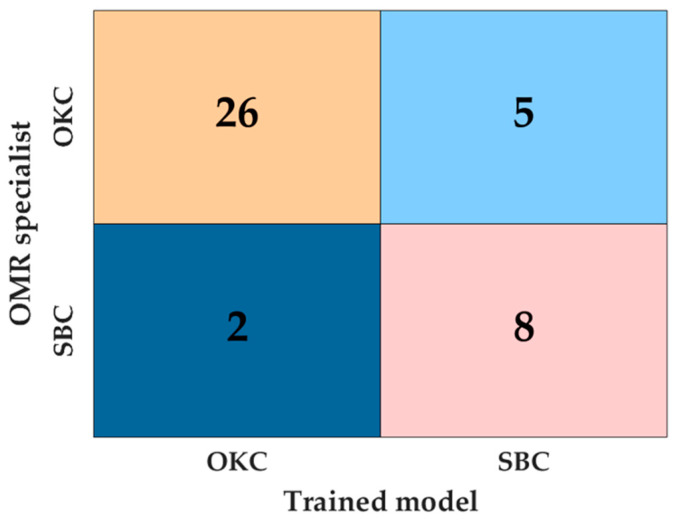
Cross-tabulation of specialists and AI diagnoses. This figure shows a cross-tabulation table comparing the diagnostic results of an oral and maxillofacial radiologist (OMR) specialist and an AI model.

**Figure 6 diagnostics-14-01144-f006:**
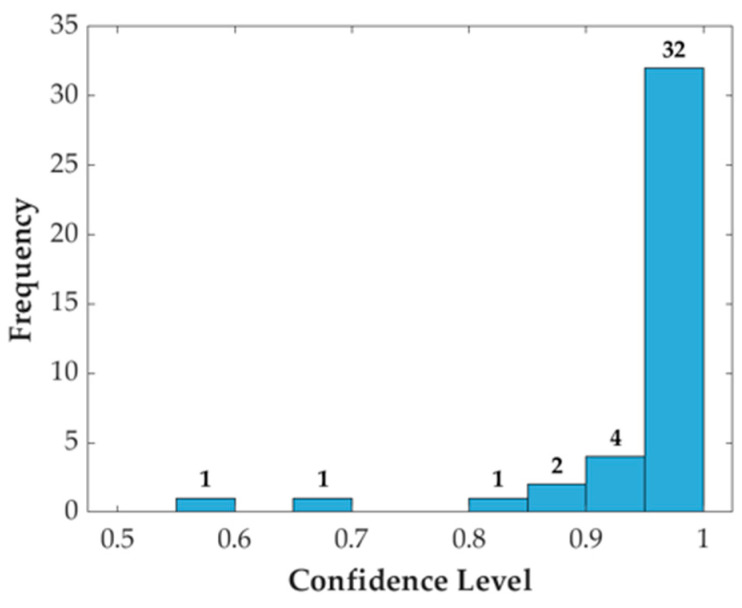
Distribution of AI confidence levels. This histogram illustrates the distribution of confidence levels for the AI model’s predictions in diagnosing OKC and SBC. The numbers above the bars indicate the actual count of predictions for each bin.

**Table 1 diagnostics-14-01144-t001:** Properties of Inception-ResNet-V2.

Network Model	Depth	Size (MB)	Parameters (Millions)	Input Image Size
Inception ResNet-V2	164	209.0	55.9	299 × 299 × 3

**Table 2 diagnostics-14-01144-t002:** Comparison of diagnostic performance between trained models, OMR * specialists, and GP **.

	Accuracy	Precision	Recall	F1 Score	Mean Time For Interpretation (Validation Data Set, Sec.)
Trained model	0.829	0.800	0.615	0.695	0.14
OMR specialist	0.805	0.727	0.615	0.667	7.19
GP					
1	0.439	0.273	0.462	0.343	7.78
2	0.707	0.529	0.692	0.600	8.48
3	0.463	0.355	0.846	0.500	13.93

* OMR: Oral and Maxillofacial Radiology Specialists, ** GP: General Practitioners.

## Data Availability

Panoramic images cannot be disclosed due to the presence of sensitive information, such as teeth and dental restoration.

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
