# Peer review of "Differential Diagnosis of OKC and SBC on Panoramic Radiographs: Leveraging Deep Learning Algorithms"

_diagnostics, 2024, doi:10.3390/diagnostics14111144_

Round 1

Reviewer 1 Report

Comments and Suggestions for Authors

Today it is very popular to apply machine learning to diagnostic questions in dentistry, reflecting that different types of radiological images play an important role in this field. Many of the papers demonstrate that some degree of diagnostic accuracy can be reached – the authors cite several examples of this type.  It is a common problem with many of these papers that they demonstrate some degree of accuracy, but that they fail to discuss whether the accuracy obtained is of clinical relevance.

This is not different with the current paper. Is an accuracy of 80% indeed “notable” as claimed by the authors?

To answer this question, two aspects have to be considered.

The first aspect is the accuracy of potential competitors. This is not addressed in the paper. It would be of interest to know the accuracy of visual inspection of the images by “standard” dentists and of visual inspection by experts as well as the accuracy of CBCT combined with visual interpretation. This would allow to judge how “notable” the result may be.

The second aspect is the clinical impact of false positive and false negative decisions. In the introduction, the authors point to the long-term clinical impact in term of the need for specific treatment in dependence on the true status. They also point to the potential need for CBCT. However, they hesitate to make any concrete suggestion how to integrate their approach into the diagnostic pipeline. Is their approach accurate enough to avoid CBCT? This would probably require a very high positive or negative predictive value of the findings from the AI approach. The confusion matrix suggests predictive values of 26/31 = 84% and 8/10 = 80%. Is this really sufficient to suggest that CBCT can be omitted? This is the question the authors should address in the discussion. (It is not convincing to suggest at the end to ignore the problem by using AI to analyse CBCT images.)

Besides this general issue of failing to discuss the main results of the paper in an adequate manner, there are also some other issues.

1.        It remains unclear, whether the authors used a histopathological examination as reference standard or whether they used the treatment decision as reference standard. The first description given by the authors raises two questions: Is it correct that the SBC cases lack a histological verification? What has happened with patients which underwent surgical excision but failed a histological verification? Later the authors claim that all cases had a histopathological diagnose.

2.        How many images were excluded because of insufficient quality? Was this a negligible number? I think it should be mentioned/emphasized in the discussion, that the accuracy of 80% refers to images of sufficient quality, and not to all patients. It would be great to see a flow diagram starting with all patients/images made during the recruitment period and showing how many patients/images were excluded for which reasons.

3.        The inclusion criteria remain unclear. What does “patients with SBC and OKC” mean? Was this known prior to entry at the hospital? Or is this the “final diagnosis”? It is important to define carefully the clinical situation of interest (Need to distinguish between SBC and OKC?) and to include all images, for which this decision was attempted during the recruitment period.

4.        The phrasing “images of cases with previously established histopathological diagnose” raises the question what has happened with those without a histopathological diagnose. Is this a selection which may influence the results?

5.        The authors claim in the abstract to use a 10% testing sample. They later include 41 out of 204 images in the testing sample, i.e. 20%.

6.        It remains unclear why the authors use cross validation, if they use at the end a test set to evaluate the diagnostic accuracy?

7.        It is great to use CAM to obtain some insights into what the AI approach is using. However, it would be of even greater interest to see examples where the AI approach failed.  

8.        The authors mentioned that both age and the vital status of teeth can be highly informative in this context. This raises three questions: What would be the accuracy if we use only age and/or vital status for decision making? How can the authors exclude that their AI approach is (partially) using indicators of age and vital status? Can we improve the accuracy by allowing the AI to know age and vital status?

Reviewer 2 Report

Comments and Suggestions for Authors

The study addresses a significant clinical problem—differentiating between odontogenic keratocyst (OKC) and simple bone cyst (SBC) using panoramic radiographs. The results are promising for the use of AI in clinical diagnostics, potentially reducing the need for more invasive procedures.

  1. Lack of Comparative Analysis: The paper would benefit from comparing the deep learning approach with traditional diagnostic methods to contextualize the improvement offered by the proposed model.

  2. Limitations and Future Work: The discussion on limitations is adequate but could be expanded to address potential biases introduced by the retrospective design and the single-center study environment. Future work could include multi-center trials and the integration of other imaging modalities.

Reviewer 3 Report

Comments and Suggestions for Authors

This paper provides a comprehensive exploration of the application of deep learning algorithms in differentiating between odontogenic keratocysts (OKC) and simple bone cysts (SBC) based on panoramic radiographs.

The inclusion of Class Activation Mapping (CAM) provides insightful visualizations, highlighting crucial regions for differentiation.

However, the paper could benefit from clearer organization and structure, especially in the methods and results sections. Providing more concise explanations and breaking down complex processes into simpler terms would enhance readability.

Additionally, while the study's findings are promising, limitations in clinical data and the use of only panoramic radiographs underscore the need for further research, potentially incorporating cone beam computed tomography (CBCT) for improved accuracy.

Round 2

Reviewer 1 Report

Comments and Suggestions for Authors

I highly appreciate the efforts of the authors to improve the manuscript. The paper is now much easier to read, the comparison allows to judge the relevance of the accuracy, and the discussion is much more nuanced and this way more adequate.

I have only three minor suggestions.

1)        It would be helpful to know a little bit more about how the OMR specialist and the GPs were selected. It is somewhat unusual to use GPs in a dental context. Were the GPs to some degree familiar with dental medicine?

2)        It would be nice to see a cross tabulation of the OMR specialist against the AI results in order to judge, whether the AI mimics the human decision process or whether it uses a different strategy.

3)        The proposal of a tiered diagnostic strategy is completely adequate. However, it would be helpful to know the distribution of the confidence level observed in this study. I suppose that the output of the algorithm provides information on this “confidence level” for each image, especially a posterior probability or predictive value. The suggestion of a tiered diagnostic strategy is only convincing, if a substantial proportion of the images resulted in a sufficient confidence level, e.g. 95%. I would hence suggest presenting also the observed distribution of these values.

Reviewer 3 Report

Comments and Suggestions for Authors

all comments / suggestions were taken by authors.

Author Response

Thank you so much